# Thompson Sampling for 1-Dimensional Exponential Family Bandits

**Nathaniel Korda**
INRIA Lille - Nord Europe, Team SequeL
nathaniel.korda@inria.fr

**Emilie Kaufmann**
Institut Mines-Telecom; Telecom ParisTech
kaufmann@telecom-paristech.fr

**Remi Munos INRIA Lille - Nord Europe, Team SequeL**
remi.munos@inria.fr

## Abstract

Thompson Sampling has been demonstrated in many complex bandit models, however the theoretical guarantees available for the parametric multi-armed bandit are still limited to the Bernoulli case. Here we extend them by proving asymptotic optimality of the algorithm using the Jeffreys prior for 1-dimensional exponential family bandits. Our proof builds on previous work, but also makes extensive use of closed forms for Kullback-Leibler divergence and Fisher information (through the Jeffreys prior) available in an exponential family. This allow us to give a finite time exponential concentration inequality for posterior distributions on exponential families that may be of interest in its own right. Moreover our analysis covers some distributions for which no optimistic algorithm has yet been proposed, including heavy-tailed exponential families.

## 1 Introduction

$K$-armed bandit problems provide an elementary model for exploration-exploitation tradeoffs found at the heart of many online learning problems. In such problems, an agent is presented with $K$ distributions (also called arms, or actions) $\{p_a\}_{a=1}^K$, from which she draws samples interpreted as rewards she wants to maximize. This objective induces a trade-off between choosing to sample a distribution that has already yielded high rewards, and choosing to sample a relatively unexplored distribution at the risk of loosing rewards in the short term. Here we make the assumption that the distributions, $p_a$, belong to a parametric family of distributions $\mathcal{P} = \{p(\cdot \mid \theta), \theta \in \Theta\}$ where $\Theta \subset \mathbb{R}$. The bandit model is described by a parameter $\theta_0 = (\theta_1, \ldots, \theta_K)$ such that $p_a = p(\cdot \mid \theta_a)$. We introduce the mean function $\mu(\theta) = \mathbb{E}_{X \sim p(\cdot|\theta)}[X]$, and the optimal arm $\theta^* = \theta_{a^*}$ where $a^* = \operatorname{argmax}_a \mu(\theta_a)$.

An algorithm, $\mathcal{A}$, for a $K$-armed bandit problem is a (possibly randomised) method for choosing which arm $a_t$ to sample from at time $t$, given a history of previous arm choices and obtained rewards, $\mathcal{H}_{t-1} := ((a_s, x_s))_{s=1}^{t-1}$: each reward $x_s$ is drawn from the distribution $p_{a_s}$. The agent's goal is to design an algorithm with low regret:

$$\mathcal{R}(\mathcal{A}, t) = \mathcal{R}(\mathcal{A}, t)(\theta) := t\mu(\theta^*) - \mathbb{E}_{\mathcal{A}}\left[\sum_{s=1}^t x_s\right].$$

This quantity measures the expected performance of algorithm $\mathcal{A}$ compared to the expected performance of an optimal algorithm given knowledge of the reward distributions, i.e. sampling always from the distribution with the highest expectation.

Since the early 2000s the "optimisim in the face of uncertainty" heuristic has been a popular approach to this problem, providing both simplicity of implementation and finite-time upper bounds on the regret (e.g. [4, 7]). However in the last two years there has been renewed interest in the Thompson Sampling heuristic (TS). While this heuristic was first put forward to solve bandit problems eighty years ago in [15], it was not until recently that theoretical analyses of its performance were achieved [1, 2, 11, 13]. In this paper we take a major step towards generalising these analyses to the same level of generality already achieved for "optimistic" algorithms.

**Thompson Sampling**  Unlike optimistic algorithms which are often based on confidence intervals, the Thompson Sampling algorithm, denoted by $\mathcal{A}_{\pi_0}$ uses Bayesian tools and puts a prior distribution $\pi_{a,0} = \pi_0$ on each parameter $\theta_a$. A posterior distribution, $\pi_{a,t}$, is then maintained according to the rewards observed in $\mathcal{H}_{t-1}$. At each time a sample $\theta_{a,t}$ is drawn from each posterior $\pi_{a,t}$ and then the algorithm chooses to sample $a_t = \arg\max_{a \in \{1,\ldots,K\}} \{\mu(\theta_{a,t})\}$. Note that actions are sampled according to their posterior probabilities of being optimal.

**Our contributions**  TS has proved to have impressive empirical performances, very close to those of state of the art algorithms such as DMED and KL-UCB [11, 9, 7]. Furthermore recent works [11, 2] have shown that in the special case where each $p_a$ is a Bernoulli distribution $\mathcal{B}(\theta_a)$, TS using a uniform prior over the arms is asymptotically optimal in the sense that it achieves the asymptotic lower bound on the regret provided by Lai and Robbins in [12] (that holds for univariate parametric bandits). As explained in [1, 2], Thompson Sampling with uniform prior for Bernoulli rewards can be slightly adapted to deal with bounded rewards. However, there is no notion of asymptotic optimality for this non-parametric family of rewards. In this paper, we extend the optimality property that holds for Bernoulli distributions to more general families of parametric rewards, namely 1-dimensional exponential families if the algorithm uses the Jeffreys prior:

**Theorem 1.** *Suppose that the reward distributions belong to a* 1*-dimensional canonical exponential family and let* $\pi_J$ *denote the associated Jeffreys prior. Then,*

$$\lim_{T \to \infty} \frac{\mathcal{R}(\mathcal{A}_{\pi_J}, T)}{\ln T} = \sum_{a=1}^{K} \frac{\mu(\theta_{a^*}) - \mu(\theta_a)}{K(\theta_a, \theta_{a^*})}, \tag{1}$$

*where* $K(\theta, \theta') := KL(p_\theta, p'_\theta)$ *is the Kullback-Leibler divergence between* $p_\theta$ *and* $p'_\theta$.

This theorem follows directly from Theorem 2. In the proof of this result we provide in Theorem 4 a finite-time, exponential concentration bound for posterior distributions of exponential family random variables, something that to the best of our knowledge is new to the literature and of interest in its own right. Our proof also exploits the connection between the Jeffreys prior, Fisher information and the Kullback-Leibler divergence in exponential families.

**Related Work**  Another line of recent work has focused on distribution-independent bounds for Thompson Sampling. [2] establishes that $\mathcal{R}(\mathcal{A}_{\pi_U}, T) = O(\sqrt{KT \ln(T)})$ for Thompson Sampling for bounded rewards (with the classic uniform prior $\pi_U$ on the underlying Bernoulli parameter). [14] go beyond the Bernoulli model, and give an upper bound on the Bayes risk (i.e. the regret averaged over the prior) independent of the prior distribution. For the parametric multi-armed bandit with $K$ arms described above, their result states that the regret of Thompson Sampling using a prior $\pi_0$ is not too big when averaged over this same prior:

$$\mathbb{E}_{\theta \sim \pi_0^{\otimes K}}[\mathcal{R}(\mathcal{A}_{\pi_0}, T)(\theta)] \leq 4 + K + 4\sqrt{KT \log(T)}.$$

Building on the same ideas, [6] have improved this upper bound to $14\sqrt{KT}$. In our paper, we rather see the prior used by Thompson Sampling as a tool, and we want therefore to derive regret bounds for any given problem parametrized by $\theta$ that depend on this parameter.

[14] also use Thompson Sampling in more general models, like the linear bandit model. Their result is a bound on the Bayes risk that does not depend on the prior, whereas [3] gives a first bound on the regret in this model. Linear bandits consider a possibly infinite number of arms whose mean rewards are linearly related by a single, unknown coefficient vector. Once again, the analysis in [3] encounters the problem of describing the concentration of posterior distributions. However by using a conjugate normal prior, they can employ explicit concentration bounds available for Normal distributions to complete their argument.

**Paper Structure**  In Section 2 we describe important features of the one-dimensional canonical exponential families we consider, including closed-form expression for KL-divergences and the Jeffreys' prior. Section 3 gives statements of the main results, and provides the proof of the regret bound. Section 4 proves the posterior concentration result used in the proof of the regret bound.

## 2   Exponential Families and the Jeffreys Prior

A distribution is said to belong to a one-dimensional canonical exponential family if it has a density with respect to some reference measure $\nu$ of the form:

$$p(x \mid \theta) = A(x) \exp(T(x)\theta - F(\theta)), \tag{2}$$

where $\theta \in \Theta \subset \mathbb{R}$. $T$ and $A$ are some fixed functions that characterize the exponential family and $F(\theta) = \log\left(\int A(x) \exp[T(x)\theta]\, d\nu(x)\right)$. $\Theta$ is called the *parameter space*, $T(x)$ the *sufficient statistic*, and $F(\theta)$ the *normalisation function*. We make the classic assumption that $F$ is twice differentiable with a continuous second derivative. It is well known [17] that:

$$\mathbb{E}_{X|\theta}(T(X)) = F'(\theta) \quad \text{and} \quad \text{Var}_{X|\theta}[T(X)] = F''(\theta)$$

showing in particular that $F$ is strictly convex. The mean function $\mu$ is differentiable and stricly increasing, since we can show that

$$\mu'(\theta) = \text{Cov}_{X|\theta}(X, T(X)) > 0.$$

In particular, this shows that $\mu$ is one-to-one in $\theta$.

**KL-divergence in Exponential Families**  In an exponential family, a direct computation shows that the Kullback-Leibler divergence can be expressed as a Bregman divergence of the normalisation function, F:

$$K(\theta, \theta') = D_F^B(\theta', \theta) := F(\theta') - [F(\theta) + F'(\theta)(\theta' - \theta)]. \tag{3}$$

**Jeffreys prior in Exponential Families**  In the Bayesian literature, a special "non-informative" prior, introduced by Jeffreys in [10], is sometimes considered. This prior, called the Jeffreys prior, is invariant under re-parametrisation of the parameter space, and it can be shown to be proportional to the square-root of the Fisher information $I(\theta)$. In the special case of the canonical exponential family, the Fisher information takes the form $I(\theta) = F''(\theta)$, hence the Jeffreys prior for the model (2) is

$$\pi_J(\theta) \propto \sqrt{|F''(\theta)|}.$$

Under the Jeffreys prior, the posterior on $\theta$ after $n$ observations is given by

$$p(\theta|y_1, \ldots y_n) \propto \sqrt{F''(\theta)} \exp\left(\theta \sum_{i=1}^{n} T(y_i) - nF(\theta_i)\right) \tag{4}$$

When $\int_\Theta \sqrt{F''(\theta)}d\theta < +\infty$, the prior is called *proper*. However, stasticians often use priors which are not proper: the prior is called *improper* if $\int_\Theta \sqrt{F''(\theta)}d\theta = +\infty$ and any observation makes the corresponding posterior (4) integrable.

**Some Intuition for choosing the Jeffreys Prior**  In the proof of our concentration result for posterior distributions (Theorem 4) it will be crucial to lower bound the prior probability of an $\epsilon$-sized KL-divergence ball around each of the parameters $\theta_a$. Since the Fisher information $F''(\theta) = \lim_{\theta' \to \theta} K(\theta, \theta')/|\theta - \theta'|^2$, choosing a prior proportional to $F''(\theta)$ ensures that the prior measure of such balls are $\Omega(\sqrt{\epsilon})$.

**Examples and Pseudocode**  Algorithm 1 presents pseudocode for Thompson Sampling with the Jeffreys prior for distributions parametrized by their natural parameter $\theta$. But as the Jeffreys prior is invariant under reparametrization, if a distribution is parametrised by some parameter $\lambda \not\equiv \theta$, the algorithm can use the Jeffreys prior $\propto \sqrt{I(\lambda)}$ on $\lambda$, drawing samples from the posterior on $\lambda$. Note that the posterior sampling step (in bold) is always tractable using, for example, a Hastings-Metropolis algorithm.

---
**Algorithm 1** Thompson Sampling for Exponential Families with the Jeffreys prior
---
**Require:** $F$ normalization function, $T$ sufficient statistic, $\mu$ mean function
  **for** $t = 1 \ldots K$ **do**
    Sample arm $t$ and get rewards $x_t$
    $N_t = 1, S_t = T(x_t)$.
  **end for**
  **for** $t = K + 1 \ldots n$ **do**
    **for** $a = 1 \ldots K$ **do**
      **Sample** $\theta_{a,t}$ **from** $\pi_{a,t} \propto \sqrt{F''(\theta)} \exp\left(\theta S_a - N_a F(\theta)\right)$
    **end for**
    Sample arm $A_t = \mathrm{argmax}_a \mu(\theta_{a,t})$ and get reward $x_t$
    $S_{A_t} = S_{A_t} + T(x_t) \quad N_{A_t} = N_{A_t} + 1$
  **end for**
---

| Name | Distribution | $\theta$ | Prior on $\lambda$ | Posterior on $\lambda$ |
|---|---|---|---|---|
| $\mathcal{B}(\lambda)$ | $\lambda^x (1-\lambda)^{1-x} \delta_{0,1}$ | $\log\left(\frac{\lambda}{1-\lambda}\right)$ | Beta $\left(\frac{1}{2}, \frac{1}{2}\right)$ | Beta $\left(\frac{1}{2} + s, \frac{1}{2} + n - s\right)$ |
| $\mathcal{N}(\lambda, \sigma^2)$ | $\frac{1}{\sqrt{2\pi\sigma^2}} e^{-\frac{(x-\lambda)^2}{2\sigma^2}}$ | $\frac{\lambda}{\sigma^2}$ | $\propto 1$ | $\mathcal{N}\left(\frac{s}{n}, \frac{\sigma^2}{n}\right)$ |
| $\Gamma(k, \lambda)$ | $\frac{\lambda^k}{\Gamma(k)} x^{k-1} e^{-\lambda x} 1_{[0,+\infty[}(x)$ | $-\lambda$ | $\propto \frac{1}{\lambda}$ | $\Gamma(kn, s)$ |
| $\mathcal{P}(\lambda)$ | $\frac{\lambda^x e^{-\lambda}}{x!} \delta_{\mathbb{N}}(x)$ | $\log(\lambda)$ | $\propto \frac{1}{\sqrt{\lambda}}$ | $\Gamma\left(\frac{1}{2} + s, n\right)$ |
| Pareto$(x_m, \lambda)$ | $\frac{\lambda x_m^\lambda}{x^{\lambda+1}} 1_{[x_m,+\infty[}(x)$ | $-\lambda - 1$ | $\propto \frac{1}{\lambda}$ | $\Gamma\left(n+1, s - n\log x_m\right)$ |
| Weibull$(k, \lambda)$ | $k\lambda(x\lambda)^{k-1} e^{-(\lambda x)^k} 1_{[0,+\infty[}$ | $-\lambda^k$ | $\propto \frac{1}{\lambda^k}$ | $\alpha\lambda^{(n-1)k} \exp(-\lambda^k s)$ |

Figure 1: The posterior distribution after observations $y_1, \ldots, y_n$ depends on $n$ and $s = \sum_{i=1}^{n} T(y_i)$

Some examples of common exponential family models are given in Figure 1, together with the posterior distributions on the parameter $\lambda$ that is used by TS with the Jeffreys prior. In addition to examples already studied in [7] for which $T(x) = x$, we also give two examples of more general canonical exponential families, namely the Pareto distribution with known min value and unknown tail index $\lambda$, Pareto$(x_m, \lambda)$, for which $T(x) = \log(x)$, and the Weibul distribution with known shape and unknown rate parameter, Weibull$(k, \lambda)$, for which $T(x) = x^k$. These last two distributions are not covered even by the work in [8], and belong to the family of heavy-tailed distributions.

For the Bernoulli model, we note futher that the use of the Jeffreys prior is not covered by the previous analyses. These analyses make an extensive use of the uniform prior, through the fact that the coefficient of the Beta posteriors they consider have to be integers.

## 3 Results and Proof of Regret Bound

An *exponential family $K$-armed bandit* is a $K$-armed bandit for which the reward distributions $p_a$ are known to be elements of an exponential family of distributions $\mathcal{P}(\Theta)$. We denote by $p_{\theta_a}$ the distribution of arm $a$ and its mean by $\mu_a = \mu(\theta_a)$.

**Theorem 2** (**Regret Bound**). *Assume that $\mu_1 > \mu_a$ for all $a \neq 1$, and that $\pi_{a,0}$ is taken to be the Jeffreys prior over $\Theta$. Then for every $\epsilon > 0$ there exists a constant $\mathcal{C}(\epsilon, \mathcal{P})$ depending on $\epsilon$ and on the problem $\mathcal{P}$ such that the regret of Thompson Sampling using the Jeffreys prior satisfies*

$$\mathcal{R}(\mathcal{A}_{\pi_J}, T) \leq \frac{1+\epsilon}{1-\epsilon}\left(\sum_{a=2}^{K} \frac{(\mu_1 - \mu_a)}{K(\theta_a, \theta_1)}\right) \ln(T) + \mathcal{C}(\epsilon, \mathcal{P}).$$

**Proof:** We give here the main argument of the proof of the regret bound, which proceed by bounding the expected number of draws of any suboptimal arm. Along the way we shall state concentration results whose proofs are postponed to later sections.

**Step 0: Notation** We denote by $y_{a,s}$ the $s$-th observation of arm $a$ and by $N_{a,t}$ the number of times arm $a$ is chosen up to time $t$. $(y_{a,s})_{s \geq 1}$ is i.i.d. with distribution $p_{\theta_a}$. Let $Y_a^u := (y_{a,s})_{1 \leq s \leq u}$ be the vector of first $u$ observations from arm $a$. $Y_{a,t} := Y_a^{N_{a,t}}$ is therefore the vector of observations from arm $a$ available at the beginning of round $t$. Recall that $\pi_{a,t}$, respectively $\pi_{a,0}$, is the posterior, respectively the prior, on $\theta_a$ at round $t$ of the algorithm.

We define $L(\theta)$ to be such that $\mathbb{P}_{Y \sim p(\cdot|\theta)}(p(Y|\theta) \geq L(\theta)) \geq \frac{1}{2}$. Observations from arm $a$ such that $p(y_{a,s}|\theta) \geq L(\theta_a)$ can therefore be seen as likely observations. For any $\delta_a > 0$, we introduce the event $\tilde{E}_{a,t} = \tilde{E}_{a,t}(\delta_a)$:

$$\tilde{E}_{a,t} = \left( \exists 1 \leq s' \leq N_{a,t} : p(y_{a,s'}|\theta_a) \geq L(\theta_a), \left| \frac{\sum_{s=1, s \neq s'}^{N_{a,t}} T(y_{a,s})}{N_{a,t} - 1} - F'(\theta_a) \right| \leq \delta_a \right). \quad (5)$$

For all $a \neq 1$ and $\Delta_a$ such that $\mu_a < \mu_a + \Delta_a < \mu_1$, we introduce
$$E_{a,t}^\theta = E_{a,t}^\theta(\Delta_a) := \left( \mu(\theta_{a,t}) \leq \mu_a + \Delta_a \right).$$

On $\tilde{E}_{a,t}$, the empirical sufficient statistic of arm $a$ at round $t$ is well concentrated around its mean and a 'likely' realization of arm $a$ has been observed. On $E_{a,t}^\theta$, the mean of the distribution with parameter $\theta_{a,t}$ does not exceed by much the true mean, $\mu_a$. $\delta_a$ and $\Delta_a$ will be carefully chosen at the end of the proof.

**Step 1: Decomposition** The idea of the proof is to decompose the probability of playing a suboptimal arm using the events given in Step 0, and that $\mathbb{E}[N_{a,T}] = \sum_{t=1}^T \mathbb{P}(a_t = a)$:

$$\mathbb{E}[N_{a,T}] = \underbrace{\sum_{t=1}^T \mathbb{P}\left( a_t = a, \tilde{E}_{a,t}, E_{a,t}^\theta \right)}_{(A)} + \underbrace{\sum_{t=1}^T \mathbb{P}\left( a_t = a, \tilde{E}_{a,t}, (E_{a,t}^\theta)^c \right)}_{(B)} + \underbrace{\sum_{t=1}^T \mathbb{P}\left( a_t = a, \tilde{E}_{a,t}^c \right)}_{(C)}.$$

where $E^c$ denotes the complement of event $E$. Term (C) is controlled by the concentration of the empirical sufficient statistic, and (B) is controlled by the tail probabilities of the posterior distribution. We give the needed concentration results in Step 2. When conditioned on the event that the optimal arm is played at least polynomially often, term (A) can be decomposed further, and then controled by the results from Step 2. Step 3 proves that the optimal arm is played this many times.

**Step 2: Concentration Results** We state here the two concentration results that are necessary to evaluate the probability of the above events.

**Lemma 3.** *Let $(y_s)$ be an i.i.d sequence of distribution $p(\cdot \mid \theta)$ and $\delta > 0$. Then*
$$\mathbb{P}\left( \left| \frac{1}{u} \sum_{s=1}^u [T(y_s) - F'(\theta)] \right| \geq \delta \right) \leq 2e^{-u\tilde{K}(\theta,\delta)},$$
*where $\tilde{K}(\theta,\delta) = \min(K(\theta + g(\delta), \theta), K(\theta - h(\delta), \theta))$, with $g(\delta) > 0$ defined by $F'(\theta + g(\delta)) = F'(\theta) + \delta$ and $h(\delta) > 0$ defined by $F'(\theta - h(\delta)) = F'(\theta) - \delta$.*

The two following inequalities that will be useful in the sequel can easily be deduced from Lemma 3. Their proof is gathered in Appendix A with that of Lemma 3. For any arm $a$, for any $b \in ]0, 1[$,

$$\sum_{t=1}^T \mathbb{P}(a_t = a, (\tilde{E}_{a,t}(\delta_a))^c) \leq \sum_{t=1}^\infty \left( \frac{1}{2} \right)^t + \sum_{t=1}^\infty 2t e^{-(t-1)\tilde{K}(\theta_a, \delta_a)} \quad (6)$$

$$\sum_{t=1}^T \mathbb{P}((\tilde{E}_{a,t}(\delta_a))^c \cap N_{a,t} > t^b) \leq \sum_{t=1}^\infty t \left( \frac{1}{2} \right)^{t^b} + \sum_{t=1}^\infty 2t^2 e^{-(t^b-1)\tilde{K}(\theta_a, \delta_a)}, \quad (7)$$

The second result tells us that concentration of the empirical sufficient statistic around its mean implies concentration of the posterior distribution around the true parameter:

**Theorem 4 (Posterior Concentration).** *Let $\pi_{a,0}$ be the Jeffreys prior. There exists constants $C_{1,a} = C_1(F, \theta_a) > 0$, $C_{2,a} = C_2(F, \theta_a, \Delta_a) > 0$, and $N(\theta_a, F)$ s.t., $\forall N_{a,t} \geq N(\theta_a, F)$,*

$$\mathbf{1}_{\tilde{E}_{a,t}} \mathbb{P}\left( \mu(\theta_{a,t}) > \mu(\theta_a) + \Delta_a | Y_{a,t} \right) \leq C_{1,a} e^{-(N_{a,t}-1)(1-\delta_a C_{2,a})K(\theta_a, \mu^{-1}(\mu_a + \Delta_a)) + \ln(N_{a,t})}$$

*whenever $\delta_a < 1$ and $\Delta_a$ are such that $1 - \delta_a C_{2,a}(\Delta_a) > 0$.*

**Step 3: Lower Bound the Number of Optimal Arm Plays with High Probability** The main difficulty adressed in previous regret analyses for Thompson Sampling is the control of the number of draws of the optimal arm. We provide this control in the form of Proposition 5 which is adapted from Proposition 1 in [11]. The proof of this result, an outline of which is given in Appendix D, explores in depth the randomised nature of Thompson Sampling. In particular, we show that the proof in [11] can be significantly simplified, but at the expense of no longer being able to describe the constant $C_b$ explicitly:

**Proposition 5.** $\forall b \in (0,1)$, $\exists C_b(\pi, \mu_1, \mu_2, K) < \infty$ such that $\sum_{t=1}^{\infty} \mathbb{P}\left(N_{1,t} \leq t^b\right) \leq C_b$.

**Step 4: Bounding the Terms of the Decomposition** Now we bound the terms of the decomposition as discussed in Step 1: An upper bound on term (C) is given in (6), whereas a bound on term (B) follows from Lemma 6 below. Although the proof of this lemma is standard, and bears a strong similarity to Lemma 3 of [3], we provide it in Appendix C for the sake of completeness.

**Lemma 6.** *For all actions $a$ and for all $\epsilon > 0$, $\exists\, N_\epsilon = N_\epsilon(\delta_a, \Delta_a, \theta_a) > 0$ such that*

$$(B) \leq [(1-\epsilon)(1-\delta_a C_{2,a})K(\theta_a, \mu^{-1}(\mu_a + \Delta_a))]^{-1} \ln(T) + \max\{N_\epsilon, N(\theta_a, F)\} + 1.$$

*where $N_\epsilon = N_\epsilon(\delta_a, \Delta_a, \theta_a)$ is the smallest integer such that for all $n \geq N_\epsilon$*

$$(n-1)^{-1} \ln(C_{1,a} n) < \epsilon(1 - \delta_a C_{2,a})K(\theta_a, \mu^{-1}(\mu_a + \Delta_a)),$$

*and $N(\theta_a, F)$ is the constant from Theorem 4.*

When we have seen enough observations on the optimal arm, term (A) also becomes a result about the concentration of the posterior and the empirical sufficient statistic, but this time for the optimal arm:

$$(A) \leq \sum_{t=1}^{T} \mathbb{P}\left(a_t = a, \tilde{E}_{a,t}, E_{a,t}^\theta, N_{1,t} > t^b\right) + C_b \leq \sum_{t=1}^{T} \mathbb{P}\left(\mu(\theta_{1,t}) \leq \mu_1 - \Delta_a', N_{1,t} > t^b\right) + C_b$$

$$\leq \underbrace{\sum_{t=1}^{T} \mathbb{P}\left(\mu(\theta_{1,t}) \leq \mu_1 - \Delta_a', \tilde{E}_{1,t}(\delta_1), N_{1,t} > t^b\right)}_{B'} + \underbrace{\sum_{t=1}^{T} \mathbb{P}\left(\tilde{E}_{1,t}^c(\delta_1) \cap N_{1,t} > t^b\right)}_{C'} + C_b \quad (8)$$

where $\Delta_a' = \mu_1 - \mu_a - \Delta_a$ and $\delta_1 > 0$ remains to be chosen. The first inequality comes from Proposition 5, and the second inequality comes from the following fact: if arm 1 is not chosen and arm $a$ is such that $\mu(\theta_{a,t}) \leq \mu_a + \Delta_a$, then $\mu(\theta_{1,t}) \leq \mu_a + \Delta_a$. A bound on term (C') is given in (7) for $a = 1$ and $\delta_1$. In Theorem 4, we bound the conditional probability that $\mu(\theta_{a,t})$ exceed the true mean. Following the same lines, we can also show that

$$\mathbb{P}\left(\mu(\theta_{1,t}) \leq \mu_1 - \Delta_a' | Y_{1,t}\right) \mathbf{1}_{\tilde{E}_{1,t}(\delta_1)} \leq C_{1,1} e^{-(N_{1,t}-1)(1-\delta_1 C_{2,1})K(\theta_1, \mu^{-1}(\mu_1 - \Delta_a')) + \ln(N_{1,t})}.$$

For any $\Delta_a' > 0$, one can choose $\delta_1$ such that $1 - \delta_1 C_{1,1} > 0$. Then, with $N = N(\mathcal{P})$ such that the function $u \mapsto e^{-(u-1)(1-\delta_1 C_{2,1})K(\theta_1, \mu^{-1}(\mu_1 - \Delta_a')) + \ln u}$ is decreasing for $u \geq N$, $(B')$ is bounded by

$$N^{1/b} + \sum_{t=N^{1/b}+1}^{\infty} C_{1,1} e^{-(t^b-1)(1-\delta_1 C_{2,1})K(\theta_1, \mu^{-1}(\mu_1 - \Delta_a')) + \ln(t^b)} < \infty.$$

**Step 4: Choosing the Values $\delta_a$ and $\epsilon_a$** So far, we have shown that for any $\epsilon > 0$ and for any choice of $\delta_a > 0$ and $0 < \Delta_a < \mu_1 - \mu_a$ such that $1 - \delta_a C_{2,a} > 0$, there exists a constant $\mathcal{C}(\delta_a, \Delta_a, \epsilon, \mathcal{P})$ such that

$$\mathbb{E}[N_{a,T}] \leq \frac{\ln(T)}{(1 - \delta_a C_{2,a})K(\theta_a, \mu^{-1}(\mu_a + \Delta_a))(1-\epsilon)} + \mathcal{C}(\delta_a, \Delta_a, \epsilon, \mathcal{P})$$

The constant is of course increasing (dramatically) when $\delta_a$ goes to zero, $\Delta_a$ to $\mu_1 - \mu_a$, or $\epsilon$ to zero. But one can choose $\Delta_a$ close enough to $\mu_1 - \mu_a$ and $\delta_a$ small enough, such that

$$(1 - C_{2,a}(\Delta_a)\delta_a)K(\theta_a, \mu^{-1}(\mu_a + \Delta_a)) \geq \frac{K(\theta_a, \theta_1)}{(1+\epsilon)},$$

and this choice leads to

$$\mathbb{E}[N_{a,T}] \leq \frac{1+\epsilon}{1-\epsilon} \frac{\ln(T)}{K(\theta_a, \theta_1)} + \mathcal{C}(\delta_a, \Delta_a, \epsilon, \mathcal{P}).$$

Using that $\mathcal{R}(\mathcal{A}, T) = \sum_{a=2}^{K}(\mu_1 - \mu_a)\mathbb{E}_{\mathcal{A}}[N_{a,T}]$ for any algorithm $\mathcal{A}$ concludes the proof. $\square$

# 4 Posterior Concentration: Proof of Theorem 4

For ease of notation, we drop the subscript $a$ and let $(y_s)$ be an i.i.d. sequence of distribution $p_\theta$, with mean $\mu = \mu(\theta)$. Furthermore, by conditioning on the value of $N_s$, it is enough to bound $\mathbf{1}_{\tilde{E}_u} \mathbb{P}(\mu(\theta_u) \geq \mu + \Delta | Y^u)$ where $Y^u = (y_s)_{1 \leq s \leq u}$ and

$$\tilde{E}_u = \left( \exists 1 \leq s' \leq u : p(y_{s'}|\theta) \geq L(\theta), \left| \frac{\sum_{s=1, s \neq s'}^u T(y_s)}{u-1} - F'(\theta) \right| \leq \delta \right).$$

**Step 1: Extracting a Kullback-Leibler Rate** The argument rests on the following Lemma, whose proof can be found in Appendix B

**Lemma 7.** *Let $\tilde{E}_u$ be the event defined by* (5)*, and introduce $\Theta_{\theta,\Delta} := \{\theta' \in \Theta : \mu(\theta') \geq \mu(\theta) + \Delta\}$. The following inequality holds:*

$$\mathbf{1}_{\tilde{E}_u} \mathbb{P}(\mu(\theta_u) \geq \mu + \Delta | Y^u) \leq \frac{\int_{\theta' \in \Theta_{\theta,\Delta}} e^{-(u-1)\left(K[\theta,\theta'] - \delta|\theta - \theta'|\right)} \pi(\theta'|y_{s'}) d\theta'}{\int_{\theta' \in \Theta} e^{-(u-1)(K[\theta,\theta'] + \delta|\theta - \theta'|)} \pi(\theta'|y_{s'}) d\theta'}, \qquad (9)$$

*with $s' = \inf\{s \in \mathbb{N} : p(y_s|\theta) \geq L(\theta)\}$.*

**Step 2: Upper bounding the numerator of (9)** We first note that on $\Theta_{\theta,\Delta}$ the leading term in the exponential is $K(\theta, \theta')$. Indeed, from (3) we know that

$$K(\theta, \theta')/|\theta - \theta'| = |F'(\theta) - (F(\theta) - F(\theta'))/(\theta - \theta')|$$

which, by strict convexity of $F$, is strictly increasing in $|\theta - \theta'|$ for any fixed $\theta$. Now since $\mu$ is one-to-one and continuous, $\Theta_{\theta,\Delta}^c$ is an interval whose interior contains $\theta$, and hence, on $\Theta_{\theta,\Delta}$,

$$\frac{K(\theta, \theta')}{|\theta - \theta'|} \geq \frac{F(\mu^{-1}(\mu + \Delta)) - F(\theta)}{\mu^{-1}(\mu + \Delta) - \theta} - F'(\theta) := (C_2(F, \theta, \Delta))^{-1} > 0.$$

So for $\delta$ such that $1 - \delta C_2 > 0$ we can bound the numerator of (9) by:

$$\int_{\theta' \in \Theta_{\theta,\Delta}} e^{-(u-1)(K(\theta,\theta') - \delta|\theta - \theta'|)} \pi(\theta'|y_{s'}) d\theta' \leq \int_{\theta' \in \Theta_{\theta,\Delta}} e^{-(u-1)K(\theta,\theta')(1-\delta C_2)} \pi(\theta'|y_{s'}) d\theta'$$

$$\leq e^{-(u-1)(1-\delta C_2)K(\theta, \mu^{-1}(\mu + \Delta))} \int_{\Theta_{\theta,\Delta}} \pi(\theta'|y_{s'}) d\theta' \leq e^{-(u-1)(1-\delta C_2)K(\theta, \mu^{-1}(\mu + \Delta))} \qquad (10)$$

where we have used that $\pi(\cdot|y_{s'})$ is a probability distribution, and that, since $\mu$ is increasing, $K(\theta, \mu^{-1}(\mu + \Delta)) = \inf_{\theta' \in \Theta_{\theta,\Delta}} K(\theta, \theta')$.

**Step 3: Lower bounding the denominator of (9)** To lower bound the denominator, we reduce the integral on the whole space $\Theta$ to a KL-ball, and use the structure of the prior to lower bound the measure of that KL-ball under the posterior obtained with the well-chosen observation $y_{s'}$. We introduce the following notation for KL balls: for any $x \in \Theta$, $\epsilon > 0$, we define

$$B_\epsilon(x) := \{\theta' \in \Theta : K(x, \theta') \leq \epsilon\}.$$

We have $\frac{K(\theta,\theta')}{(\theta - \theta')^2} \to F''(\theta) \neq 0$ (since $F$ is strictly convex). Therefore, there exists $N_1(\theta, F)$ such that for $u \geq N_1(\theta, F)$, on $B_{\frac{1}{u^2}}(\theta)$,

$$|\theta - \theta'| \leq \sqrt{2K(\theta, \theta')/F''(\theta)}.$$

Using this inequality we can then bound the denominator of (9) whenever $u \geq N_1(\theta, F)$ and $\delta < 1$:

$$\int_{\theta' \in \Theta} e^{-(u-1)(K(\theta,\theta') + \delta|\theta - \theta'|)} \pi(\theta'|y_{s'}) d\theta' \geq \int_{\theta' \in B_{1/u^2}(\theta)} e^{-(u-1)(K(\theta,\theta') + \delta|\theta - \theta'|)} \pi(\theta'|y_{s'}) d\theta'$$

$$\geq \int_{\theta' \in B_{1/u^2}(\theta)} e^{-(u-1)\left(K(\theta,\theta') + \delta\sqrt{\frac{2K(\theta,\theta')}{F''(\theta)}}\right)} \pi(\theta'|y_{s'}) d\theta' \geq \pi\left(B_{1/u^2}(\theta)|y_{s'}\right) e^{-\left(1 + \sqrt{\frac{2}{F''(\theta)}}\right)}.$$

$$(11)$$

Finally we turn our attention to the quantity

$$\pi\left(B_{1/u^2}(\theta)|y_{s'}\right) = \frac{\int_{B_{1/u^2}(\theta)} p(y'_s|\theta')\pi_0(\theta')d\theta'}{\int_{\Theta} p(y'_s|\theta')\pi_0(\theta')d\theta'} = \frac{\int_{B_{1/u^2}(\theta)} p(y'_s|\theta')\sqrt{F''(\theta')}d\theta'}{\int_{\Theta} p(y'_s|\theta')\sqrt{F''(\theta')}d\theta'}. \qquad (12)$$

Now since the KL divergence is convex in the second argument, we can write $B_{1/u^2}(\theta) = (a, b)$. So, from the convexity of $F$ we deduce that

$$\frac{1}{u^2} = K(\theta, b) = F(b) - [F(\theta) + (b-\theta)F'(\theta)] = (b-\theta)\left[\frac{F(b) - F(\theta)}{(b-\theta)} - F'(\theta)\right]$$

$$\leq (b-\theta)\left[F'(b) - F'(\theta)\right] \leq (b-a)\left[F'(b) - F'(\theta)\right] \leq (b-a)\left[F'(b) - F'(a)\right].$$

As $p(y \mid \theta) \to 0$ as $y \to \pm\infty$, the set $\mathcal{C}(\theta) = \{y : p(y \mid \theta) \geq L(\theta)\}$ is compact. The map $y \mapsto \int_{\Theta} p(y|\theta')\sqrt{F''(\theta')}d\theta' < \infty$ is continuous on the compact $\mathcal{C}(\theta)$. Thus, it follows that

$$L'(\theta) = L'(\theta, F) := \sup_{y:p(y|\theta)>L(\theta)}\left\{\int_{\Theta} p(y|\theta')\sqrt{F''(\theta')}d\theta'\right\} < \infty$$

is an upper bound on the denominator of (12).

Now by the continuity of $F''$, and the continuity of $(y, \theta) \mapsto p(y|\theta)$ in both coordinates, there exists an $N_2(\theta, F)$ such that for all $u \geq N_2(\theta, F)$

$$F''(\theta) \geq \frac{1}{2}\frac{F'(b) - F'(a)}{b-a} \text{ and } \left(p(y|\theta')\sqrt{F''(\theta')} \geq \frac{L(\theta)}{2}\sqrt{F''(\theta)}, \forall \theta' \in B_{1/u^2}(\theta), y \in \mathcal{C}(\theta)\right).$$

Finally, for $u \geq N_2(\theta, F)$, we have a lower bound on the numerator of (12):

$$\int_{B_{1/u^2}(\theta)} p(y'_s|\theta')\sqrt{F''(\theta')}d\theta' \geq \frac{L(\theta)}{2}\sqrt{F''(\theta)}\int_a^b d\theta' = \frac{L(\theta)}{2}\sqrt{(F'(b) - F'(a))(b-a)} \geq \frac{L(\theta)}{2u}$$

**Puting everything together, we get** that there exist constants $C_2 = C_2(F, \theta, \Delta)$ and $N(\theta, F) = \max\{N_1, N_2\}$ such that for every $\delta < 1$ satisfying $1 - \delta C_2 > 0$, and for every $u \geq N$, one has

$$\mathbf{1}_{\tilde{E}_u}\mathbb{P}(\mu(\theta_u) \geq \mu(\theta) + \Delta|Y_u) \leq \frac{2e^{1+\sqrt{\frac{2}{F''(\theta)}}}L'(\theta)u}{L(\theta)}e^{-(u-1)(1-\delta C_2)K(\theta, \mu^{-1}(\mu+\Delta))}.$$

**Remark 8.** *Note that when the prior is proper we do not need to introduce the observation $y_{s'}$, which significantly simplifies the argument. Indeed in this case, in (10) we can use $\pi_0$ in place of $\pi(\cdot|y_{s'})$ which is already a probability distribution. In particular, the quantity (12) is replaced by $\pi_0\left(B_{1/u^2}(\theta)\right)$, and so the constants $L$ and $L'$ are not needed.*

## 5 Conclusion

We have shown that choosing to use the Jeffreys prior in Thompson Sampling leads to an asymptotically optimal algorithm for bandit models whose rewards belong to a 1-dimensional canonical exponential family. The cornerstone of our proof is a finite time concentration bound for posterior distributions in exponential families, which, to the best of our knowledge, is new to the literature. With this result we built on previous analyses and avoided Bernoulli-specific arguments. Thompson Sampling with Jeffreys prior is now a provably competitive alternative to KL-UCB for exponential family bandits. Moreover our proof holds for slightly more general problems than those for which KL-UCB is provably optimal, including some heavy-tailed exponential family bandits.

Our arguments are potentially generalisable. Notably generalising to $n$-dimensional exponential family bandits requires only generalising Lemma 3 and Step 3 in the proof of Theorem 4. Our result is asymptotic, but the only stage where the constants are not explicitly derivable from knowledge of $F$, $T$, and $\theta_0$ is in Lemma 9. Future work will investigate these open problems. Another possible future direction lies the optimal choice of prior distribution. Our theoretical guarantees only hold for Jeffreys' prior, but a careful examination of our proof shows that the important property is to have, for every $\theta_a$,

$$-\ln\left(\int_{(\theta':K(\theta_a, \theta')\leq n^{-2})}\pi_0(\theta')d\theta'\right) = o(n),$$

which could hold for prior distributions other than the Jeffreys prior.

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
