[Supplementary Material]

# A  Concentration of the Sufficient Statistics: Proof of Lemma 3, and Inequalities (6) and (7)

*Proof of Lemma 3.* The proof of Lemma 3 follows from the classical Cramér-Chenoff technique (see [5]). For any $\lambda > 0$.

$$A := \mathbb{P}\left(\frac{1}{u}\sum_{i=1}^{u}[T(y_i) - F'(\theta)] \geq \delta\right) = \mathbb{P}\left(e^{\lambda\left(\sum_{i=1}^{u}[T(y_i)-F'(\theta)]\right)} \geq e^{\lambda u \delta}\right)$$

$$\leq e^{-\lambda u \delta}\mathbb{E}\left[e^{\lambda\left(\sum_{i=1}^{u}[T(y_i)-F'(\theta)]\right)}\right] = e^{-u(\delta\lambda - \phi_a(\lambda))}$$

where we have used the Markov inequality, and where

$$\phi_a(\lambda) := \ln \mathbb{E}_{X|\theta}\left[e^{\lambda(T(X)-F'(\theta))}\right] = F(\theta + \lambda) - F(\theta) - \lambda F'(\theta).$$

Now we optimize in $\lambda$ by choosing $\lambda > 0$ that maximizes

$$\delta\lambda - \phi_a(\lambda) = \lambda(\delta + F'(\theta)) - F(\theta + \lambda) + F(\theta) := f(\lambda).$$

$f(\lambda)$ is differentiable in $\lambda$ and its minimum, $\lambda^*$, satisfies $f'(\lambda^*) = 0$ i.e.

$$F'(\theta + \lambda^*) = \delta + F'(\theta).$$

(Note that $\lambda^* > 0$ since $F'$ is increasing). Finally, we get

$$A \leq e^{-u((\delta+F'(\theta))\lambda^* - F(\theta+\lambda^*)+F(\theta))} = e^{-u(F'(\theta+\lambda^*)\lambda^* - F(\theta+\lambda^*)+F(\theta))} = e^{-uK(\theta+\lambda^*,\theta)}.$$

The same reasoning leads to the upper bound

$$\mathbb{P}\left(\frac{1}{u}\sum_{s=1}^{u}[T(y_s) - F'(\theta)] \leq -\delta\right) \leq e^{-uK(\theta-\nu^*,\theta)},$$

where $\nu^*$ is such that $F'(\theta - \nu^*) = F'(\theta) - \delta$. $\qquad\square$

For the proof of inequalities (6) and (7), we intoduce the notation $Y_{a,s'}^{u} = Y_a^s\backslash\{y_{a,s}\}$ (the first $u$ observations of arms $a$ exept observation $y_{a,s'}$). First note that we have $\tilde{E}_{a,t}^c \subseteq B_{a,N_{a,t}} \bigcup D_{a,N_{a,t}}$, with

$$B_{a,s} = \left(\forall s' \in [1,s], p(y_{a,s'}|\theta_a) \leq L(\theta_a)\right),$$

$$D_{a,s} = \left(\exists s' \in \{1,\ldots s\} : \left|\frac{1}{s-1}\sum_{k=1,k\neq s'}^{s}(T(y_{a,k}) - F'(\theta_a))\right| \geq \delta_a\right).$$

Indeed, we have used that for two sequences of event $F_{s'}$ and $G_{s'}$,

$$\left(\bigcup_{s'=1}^{s} F_{s'} \cap G_{s'}\right)^c = \bigcap_{s'\leq s} F_{s'}^c \cup G_{s'}^c \subset \bigcap_{s'\leq s} F_{s'}^c \cup \left(\bigcup_{s''\leq s} G_{s''}^c\right) = \left(\bigcap_{s'\leq s} F_{s'}^c\right) \cup \left(\bigcup_{s'\leq s} G_{s'}^c\right).$$

One then has

$$
\begin{aligned}
\sum_{t=1}^{T}\mathbb{P}(a_t = a, \tilde{E}_{a,t}^c(\delta)) &\leq \mathbb{E}\left[\sum_{t=1}^{T}\sum_{s=1}^{t}\mathbf{1}_{(a_t=a,N_{a,t}=s)}(\mathbf{1}_{B_{a,s}} + \mathbf{1}_{D_{a,s}})\right] \\
&\leq \mathbb{E}\left[\sum_{s=1}^{T}\mathbf{1}_{B_{a,s}}\right] + \mathbb{E}\left[\sum_{s=1}^{T}\mathbf{1}_{D_{a,s}}\right] \\
&\leq \sum_{s=1}^{T}\mathbb{P}\left(p(y_{a,1}|\theta_a) \leq L(\theta_a)\right)^s + \sum_{s=1}^{T}\sum_{s'=1}^{s}\mathbb{P}\left(\left|\frac{1}{s-1}\sum_{k=1,k\neq s'}^{s}(T(y_{a,k}) - F'(\theta_a))\right| \geq \delta_a\right) \\
&\leq \sum_{s=1}^{\infty}\left(\frac{1}{2}\right)^s + \sum_{s=1}^{\infty}se^{-(s-1)\tilde{K}(\theta_a,\delta_a)},
\end{aligned}
$$

where we use that the definition of $L(\theta)$ gives $\mathbb{P}\left(p(y_{a,1}|\theta_a) \leq L(\theta_a)\right) \leq \frac{1}{2}$. This leads to inequality (6). To proof (7), we write:

$$
\begin{aligned}
\sum_{t=1}^{T} \mathbb{P}(\tilde{E}_{a,t}(\delta_a)^c \cap N_{a,t} > t^b) &\leq \mathbb{E}\left[\sum_{t=1}^{T}\sum_{s=t^b}^{t} \mathbf{1}_{N_{a,t}=s}(\mathbf{1}_{B_{a,s}} + \mathbf{1}_{D_{a,s}})\right] \\
&\leq \sum_{t=1}^{T}\sum_{s=t^b}^{t} \mathbb{P}(p(y_{a,1}|\theta_a) \leq L(\theta_a))^s \\
&\quad + \sum_{t=1}^{T}\sum_{s=t^b}^{t}\sum_{s'=1}^{s} \mathbb{P}\left(\left|\frac{1}{s-1}\sum_{k=1,k\neq s'}^{s}(T(y_{a,k}) - F'(\theta_a))\right| \geq \delta_a\right) \\
&\leq \sum_{t=1}^{T} t\left(\frac{1}{2}\right)^{t^b} + \sum_{t=1}^{T} t^2 \exp(-t^b \tilde{K}(\theta_a, \delta)).
\end{aligned}
$$

# B    Extracting the KL-divergence: Proof of Lemma 7

We assume that the event $\tilde{E}_u$ holds, $s' \leq u$. So, on this event we have

$$
\begin{aligned}
\mathbb{P}\left(\mu(\theta_u) \geq \mu + \Delta | Y^u\right) &= \frac{\int_{\theta' \in \Theta_{\theta,\Delta}} \prod_{s=1,s\neq s'}^{u} p(y_s \mid \theta')p(y_{s'}|\theta')\pi(\theta')d\theta'}{\int_{\theta' \in \Theta} \prod_{s=1,s\neq s'}^{u} p(y_s \mid \theta')p(y_{s'}|\theta')\pi(\theta')d\theta'} \\
&= \frac{\int_{\theta' \in \Theta_{\theta,\Delta}} \prod_{s=1,s\neq s'}^{u} \frac{p(y_s|\theta')}{p(y_s|\theta)}p(y_{s'}|\theta')\pi(\theta')d\theta'}{\int_{\theta' \in \Theta} \prod_{s=1,s\neq s'}^{u} \frac{p(y_s|\theta')}{p(y_s|\theta)}p(y_{s'}|\theta')\pi(\theta')d\theta'} \\
&= \frac{\int_{\theta' \in \Theta_{\theta,\Delta}} e^{-(u-1)K[Y'^u,\theta,\theta']}\pi(\theta'|y_{s'})d\theta'}{\int_{\theta' \in \Theta} e^{-(u-1)K[Y'^u,\theta,\theta']}\pi(\theta'|y_{s'})d\theta'}
\end{aligned}
$$

where $\pi(\theta|y_{s'})$ denotes the posterior distribution on $\theta$ after observation $y_{s'}$ and

$$
K[Y_{s'}^u, \theta, \theta'] := \frac{1}{u-1}\sum_{s=1,s\neq s'}^{u} \ln\frac{p(y_s \mid \theta)}{p(y_s \mid \theta')}
$$

denotes the empirical KL-divergence obtained from the observations $Y_{s'}^u = Y^u \setminus \{y_{s'}\}$. Introducing

$$
r(Y_{s'}^u, \theta') = K[Y_{s'}^u, \theta, \theta'] - \mathbb{E}_{X|\theta}\left(\ln\frac{p(X \mid \theta)}{p(X \mid \theta')}\right),
$$

we can rewrite

$$
\mathbb{P}\left(\mu(\theta_u) \geq \mu + \Delta | Y^u\right) = \frac{\int_{\theta' \in \Theta_{\theta,\Delta}} e^{-(u-1)\left(K[\theta,\theta']+r(Y'^u,\theta')\right)}\pi(\theta'|y_{s'})d\theta'}{\int_{\theta' \in \Theta} e^{-(u-1)(K[\theta,\theta']+r(Y'^u,\theta'))}\pi(\theta'|y_{s'})d\theta'}.
$$

Now, a direct computation show that

$$
|r(Y'^u, \theta')| \leq |\theta - \theta'|\left|\frac{1}{u-1}\sum_{s=1,s\neq s'}^{u}[T(y_s) - F'(\theta)]\right|. \tag{13}
$$

Indeed, for any $\theta, \theta' \in \Theta$

$$
\ln\frac{p(y \mid \theta)}{p(y \mid \theta')} = T(y)(\theta - \theta') - [F(\theta) - F(\theta')],
$$

and one also recalls that

$$K(\theta, \theta') = F'(\theta)(\theta - \theta') - [F(\theta) - F(\theta')]. \qquad (14)$$

Hence

$$|r(Y_{s'}^u, \theta, \theta')| = \left| \frac{1}{u-1} \sum_{s=1, s \neq s'}^{u} \left[ \ln \frac{p(y_s \mid \theta)}{p(y_s \mid \theta')} - K(\theta, \theta') \right] \right|$$

$$= \left| \frac{1}{u-1} \sum_{s=1, s \neq s'}^{u} [(T(x) - F'(\theta))(\theta - \theta')] \right| \leq \left| \frac{1}{u-1} \sum_{s=1, s \neq s'}^{u} [T(y_s) - \nabla F(\theta)] \right| |\theta' - \theta|.$$

The inequality (13) leads to the result, using that on $\tilde{E}_u$,

$$\left| \frac{1}{u-1} \sum_{s=1, s \neq s'}^{u} [T(y_s) - F'(\theta)] \right| \leq \delta$$

## C  Proof of Lemma 6

From Theorem 4 we know that, for $N_{a,t} \geq N(\theta_a, F)$,

$$\mathbf{1}_{\tilde{E}_{a,t}} \mathbb{P}((E_{a,t}^\theta)^c \mid \mathcal{F}_t) = \mathbf{1}_{\tilde{E}_{a,t}} \mathbb{P}((E_{a,t}^\theta)^c \mid Y_{a,t})$$

$$\leq C_{1,a} e^{-(N_{a,t}-1)(1-\delta_a C_{2,a})K(\theta_a, \mu^{-1}(\mu_a + \Delta_a)) + \ln N_{a,t}}$$

$$\leq e^{-(N_{a,t}-1)\left((1-\delta_a C_{2,a})K(\theta_a, \mu^{-1}(\mu_a + \Delta_a)) - \ln(C_{1,a} N_{a,t})/(N_{a,t}-1)\right)}$$

Let $N_\epsilon = N_\epsilon(\delta_a, \Delta_a, \theta_a)$ be the smallest integer such that for all $n \geq N_\epsilon$

$$\frac{\ln(C_{1,a} n)}{n-1} < \epsilon(1 - \delta_a C_{2,a})K(\theta_a, \mu^{-1}(\mu_a + \Delta_a)).$$

Defining

$$L_T := \frac{\ln T}{(1-\epsilon)(1 - \delta_a C_{2,a})K(\theta_a, \mu^{-1}(\mu_a + \Delta_a))}$$

we have that for all $t$ and $T$ such that $N_{a,t} - 1 \geq \max(L_T, N_\epsilon, N(\theta_a, F))$,

$$\mathbf{1}_{\tilde{E}_{a,t}} \mathbb{P}(\mu(\theta_a(t)) > \mu(\theta_a) + \Delta_a \mid \mathcal{F}_t) \leq \frac{1}{T}.$$

Let $\tau = \inf\{t \in \mathbb{N} \mid N_{a,t} \geq \max(L_T, N_\epsilon, N(\theta_a, F)) + 1\}$. $\tau$ is a stopping time with respect to $\mathcal{F}_t$. Then,

$$\sum_{t=1}^{T} \mathbb{P}\left(a_t = a, (E_{a,t}^\theta)^c, \tilde{E}_{a,t}\right) \leq \mathbb{E}\left[\sum_{t=1}^{\tau} \mathbf{1}_{(a_t=a)}\right] + \mathbb{E}\left[\sum_{t=\tau+1}^{T} \mathbf{1}_{(a_t=a)} \mathbf{1}_{\tilde{E}_{a,t}} \mathbf{1}_{(E_{a,t}^\theta)^c}\right]$$

$$= \mathbb{E}[N_{a,\tau}] + \mathbb{E}\left[\sum_{t=\tau+1}^{T} \mathbf{1}_{(a_t=a)} \mathbf{1}_{\tilde{E}_{a,t}} \mathbb{P}\left((E_{a,t}^\theta)^c \mid \mathcal{F}_t\right)\right]$$

$$= \mathbb{E}[N_{a,\tau}] + \mathbb{E}\left[\sum_{t=\tau+1}^{T} \mathbf{1}_{(a_t=a)} \mathbf{1}_{\tilde{E}_{a,t}} \mathbb{P}\left(\mu(\theta_a(t)) > \mu(\theta_a) + \Delta_a \mid Y_{a,t}\right)\right]$$

$$\leq L_T + 1 + \max(N_\epsilon, N(\theta_a, F)) + \mathbb{E}\left[\sum_{t=\tau+1}^{T} \frac{1}{T}\right]$$

$$\leq L_T + \max(N_\epsilon, N(\theta_a, F)) + 2.$$

# D  Controlling the Number of Optimal Plays: Outline Proof of Proposition 5

The proof of this proposition is quite detailed, and essentially the same as the proof given for Proposition 1 in [11], which we will sometimes refer to. However, in generalising to the case of exponential family bandits we show how to avoid the need to explicity calculate posterior probabilities that lead to Lemma 4 in [11]. While simplifying the proof we loose the ability to specify the constants explicitly, and so the analysis becomes asymptotic, but holds for every $b \in ]0, 1[$.

**Sketch of the proof and key results**   Let $\tau_j$ be the occurrence of the $j^{th}$ play of the optimal arm (with $\tau_0 := 0$). Let $\xi_j := (\tau_{j+1} - 1) - \tau_j$: this random variable measures the number of time steps between the $j^{th}$ and the $(j + 1)^{th}$ play of the optimal arm, and so $\sum_{a=2}^{K} N_{a,t} = \sum_{j=0}^{N_{1,t}} \xi_j$. We then upper bound $\mathbb{P}(N_{1,t} \leq t^b)$ as in [11]:

$$\mathbb{P}(N_{1,t} \leq t^b) \leq \mathbb{P}\left(\exists j \in \left\{0, .., \lfloor t^b \rfloor\right\} : \xi_j \geq t^{1-b} - 1\right) \leq \sum_{j=0}^{\lfloor t^b \rfloor} \mathbb{P}(\underbrace{\xi_j \geq t^{1-b} - 1}_{:= \mathcal{E}_j}) \tag{15}$$

We introduce the interval $\mathcal{I}_j = \{\tau_j, \tau_j + \lceil t^{1-b} - 1 \rceil\}$: on the event $\mathcal{E}_j$, $\mathcal{I}_j$ is included in $\{\tau_j, \tau_{j+1}\}$ and no draw of arm 1 occurs on $\mathcal{I}$. We also introduce for each arm $a \neq 1$ $d_a := \frac{\mu_1 - \mu_a}{2}$.

The idea of the rest of the analysis is based on the following remark. If on a subinterval $\mathcal{I} \subseteq [\tau_j, \tau_{j+1}[$ of size $f(t)$ arm 1 is not drawn and all the samples of the suboptimal arms fall below $\mu_2 + d_2 < \mu_1$, then for all $s \in \mathcal{I}$, $\mu(\theta_{1,s}) \leq \mu_2 + d_2$. On $\mathcal{I}$, the sequence $(\theta_{1,s})$ is i.i.d. with distribution $\pi_{1,\tau_j}$, and hence,

$$\mathbb{P}(\forall s \in \mathcal{I}, \ \mu(\theta_{1,s}) \leq \mu_2 + \delta) \leq \left(\mathbb{P}\left(\mu(\theta_{1,\tau_j}) \leq \mu_2 + \delta_2\right)\right)^{f(t)}$$

At this point, an asymptotic result, telling that the posterior on $\theta_1$ concentrates to a Dirac in $\theta_1$ (the Bernstein-Von-Mises theorem, see [16]) , leads to

$$\mathbb{P}(\mu(\theta_{1,\tau_j}) \leq \mu_2 + \delta_2) \underset{j \to \infty}{\to} 0.$$

Assuming that $\forall j, \mathbb{P}(\mu(\theta_{1,\tau_j}) \leq \mu_2 + \delta_2) \neq 1$, we have shown the following Lemma, which plays the role of an asymptotic couterpart for Lemma 3 in [11].

**Lemma 9.** *There exists a constant $C = C(\pi_0) < 1$, such that for every (random) interval $\mathcal{I}$ included in $\mathcal{I}_j$ and for every positive function $f$, one has*

$$\mathbb{P}\left(\forall s \in \mathcal{I}, \ \mu(\theta_{1,s}) \leq \mu_2 + \delta_2, \quad |\mathcal{I}| \geq f(t)\right) \leq C^{f(t)}.$$

Another key lemma is the following which generalizes Lemma 4 in [11]. The proof of this lemma is standard: it proceeds by conditioning on the event $\tilde{E}_{a,t}$[1] and applying Theorem 4, and Lemma 3.

**Lemma 10.** *For every $a \in A$, $\delta > 0$, there exist constants $C_a = C_a(\mu_a, \delta, F)$ and $N$ such that for $t \geq N$,*

$$\mathbb{P}\left(\exists s \leq t, \exists a \neq 1 : \mu(\theta_{a,s}) > \mu_a + d_a, N_{a,s} > C_a \ln(t)\right) \leq \frac{2(K - 1)}{t^2}.$$

The rest of the proof proceeds by finding a subinterval of $\mathcal{I}_j$ on which all the samples of all the suboptimal arms indeed fall below the corresponding thresholds $\mu_a + d_a$. This is done exactly as in [11] and we recall the main steps of the proof below. Before that, we need to introduce the notion of *saturated*, suboptimal action.

**Definition 11.** *Let $t$ be fixed. For any $a \neq 1$, an action $a$ is said to be* saturated *at time $s$ if it has been chosen at least $C_a \ln(t)$ times, i.e. $N_{a,t} \geq C_a \ln(t)$. We shall say that it is* unsaturated *otherwise. Furthermore at any time we call a choice of an unsaturated, suboptimal action an* interruption.

**Step 1: Decomposition of $\mathcal{I}_j$**   We want to study the process of saturation on the event $\mathcal{E}_j = \{\xi_j \geq t^{1-b} - 1\}$. We start by decomposing the interval $\mathcal{I}_j = \{\tau_j, \tau_j + \lceil t^{1-b} - 1\rceil\}$ into $K$ subintervals:

$$\mathcal{I}_{j,l} := \left\{\tau_j + \left\lceil \frac{(l-1)(t^{1-b}-1)}{K} \right\rceil, \tau_j + \left\lceil \frac{l(t^{1-b}-1)}{K} \right\rceil \right\}, \, l = 1, \ldots, K.$$

Now for each interval $\mathcal{I}_{j,l}$, we introduce:

- $\mathcal{F}_{j,l}$: the event that by the end of the interval $\mathcal{I}_{j,l}$ at least $l$ suboptimal actions are saturated;
- $n_{j,l}$: the number of interruptions during this interval.

We use the following decomposition to bound the probability of the event $\mathcal{E}_j$:

$$\mathbb{P}(\mathcal{E}_j) = \mathbb{P}(\mathcal{E}_j \cap \mathcal{F}_{j,K-1}) + \mathbb{P}(\mathcal{E}_j \cap \mathcal{F}_{j,K-1}^c) \tag{16}$$

Note that the quantities $\mathcal{E}_j$, $\mathcal{I}_{j,l}$, $\mathcal{F}_{j,l}$ and $n_{j,l}$ all depend on $t$, however we suppress this dependency for notational convenience. However, we keep in mind that we bound the different probabilities for $t \geq N$, so that Lemma 10 applies.

**Step 2: Bounding $\mathbb{P}(\mathcal{E}_j \cap \mathcal{F}_{j,K-1})$**   On the event $\mathcal{E}_j \cap \mathcal{F}_{j,K-1}$, only saturated suboptimal arms are drawn on the interval $\mathcal{I}_{j,K}$. Using Lemma 10, we get

$$
\begin{aligned}
\mathbb{P}(\mathcal{E}_j \cap \mathcal{F}_{j,K-1}) \leq &\mathbb{P}(\{\exists s \in \mathcal{I}_{j,K}, a \neq 1 : \mu(\theta_{a,s}) > \mu_a + d_a\} \cap \mathcal{E}_j \cap \mathcal{F}_{j,K-1}) \\
&+ \mathbb{P}(\{\forall s \in \mathcal{I}_{j,K}, a \neq 1 : \mu(\theta_{a,s}) \leq \mu_a + d_a\} \cap \mathcal{E}_j \cap \mathcal{F}_{j,K-1}) \\
\leq &\mathbb{P}(\exists s \leq t, a \neq 1 : \mu(\theta_{a,s}) > \mu_a + d_a, N_{a,t} > C_a \ln(t)) \\
&+ \mathbb{P}(\{\forall s \in \mathcal{I}_{j,K}, a \neq 1 : \mu(\theta_{a,s}) > \mu_a + d_a\} \cap \mathcal{E}_j \cap \mathcal{F}_{j,K-1}) \\
\leq &\frac{2(K-1)}{t^2} + \mathbb{P}(\{\forall s \in \mathcal{I}_{j,K} : \mu(\theta_{1,s}) \leq \mu_2 + d_2\} \cap \mathcal{E}_j) \\
\leq &\frac{2(K-1)}{t^2} + C^{\frac{t^{1-b}-1}{K}}.
\end{aligned}
$$

for $0 < C < 1$ as in Lemma 9. The second last inequality comes from the fact that if arm 1 is not drawn, the sample $\theta_{1,s}$ must be smaller than some sample $\theta_{a,s}$ and therefore smaller than $\mu_2 + d_2$.

**Step 3: Bounding $\mathbb{P}(\mathcal{E}_j \cap \mathcal{F}_{j,K-1}^c)$**   A similar argument to that employed in Step 2 can be used in an induction to show that for all $2 \leq l \leq K$, if $t$ is larger than some deterministic constant $N_{\mu_1,\mu_2,b}$ specified in the base case,

$$\mathbb{P}(\mathcal{E}_j \cap \mathcal{F}_{j,l-1}^c) \leq (l-2)\left(\frac{2(K-1)}{t^2} + C^{\frac{t^{1-b}-1}{CK^2 \ln(t)}}\right)$$

We refer the reader to [11] for a precise description of the induction. For $l = K$ we then get

$$\mathbb{P}(\mathcal{E}_j \cap \mathcal{F}_{j,K-1}^c) \leq (K-2)\left(\frac{2(K-1)}{t^2} + C^{\frac{t^{1-b}-1}{CK^2 \ln(t)}}\right). \tag{17}$$

**Step 4: Conclusion**   Putting Steps 2 and 3 together we obtain that for $t \geq N_0 := \max(N, N_{\mu_1,\mu_2,b})$,

$$\mathbb{P}(\mathcal{E}_j(t)) \leq \frac{2(K-1)^2}{t^2} + C^{\frac{t^{1-b}-1}{K}} + (K-2)KC\ln(t)C^{\frac{t^{1-b}-1}{CK^2 \ln(t)}},$$

$$\mathbb{P}(N_{1,t} \leq t^b) \leq \frac{2(K-1)^2}{t^{2-b}} + t^b C^{\frac{t^{1-b}-1}{K}} + (K-2)KCt^b \ln(t)C^{\frac{t^{1-b}-1}{CK^2 \ln(t)}},$$

where we use 15. It then follows that

$$\sum_{t=1}^{\infty} \mathbb{P}(N_{1,t} \leq t^b) \leq N_0 + \sum_{t=N_0+1}^{\infty} \mathbb{P}(\mathcal{E}_j) = C_b = C_b(\pi_0, \mu_1, \mu_2, K) < \infty.$$

## Footnotes

[1] Using $\tilde{E}_{a,t}$ in place of $E_{a,t}$ from [11] only changes slightly the constant $C_a$.