[Reviews · NeurIPS 2013]

Submitted by Assigned_Reviewer_3

This paper provides a regret analysis for Thompson Sampling algorithm, when Jefferys priors are used, and the reward is assumed to be generated from an exponential family of distributions. They show that TS achieves a regret upper bound equal to the optimal asymptotic lower bound on regret for the multi-armed bandit problem.

The author claims to generalize the analysis of TS by extending it to exponential family of distributions, and claims that the earlier work provided bounds only when the rewards are generated by a Bernoulli distribution. However, the latter claim is not true. The work [1,2] provided bounds for rewards generated by "any distribution" as long as the distribution has a bounded support (see Algorithm 2 in [1] and the remarks below). By the same argument even the analysis of [10] applies to the general stochastic bandits with bounded support. The author should rectify these claims made in the introduction and elsewhere in the paper.

However, the above-mentioned papers only match the Lai-Robbins lower bound in case of binary rewards. In particular, these papers provide regret bound for any bounded support distribution, so it is not possible to match the KL-divergence term in the lower bound which is distribution specific. The author of this paper can be credited to prove an optimal upper bound, which matches the Lai and Robbins lower bound in non-binary case of exponential families, which is also an unbounded support distribution.

While the analysis in the paper looks correct, I did not find any novel technique introduced in the paper. The technique follows very closely that of [10], with only technical changes. The posterior concentration for exponential families is new and might be interesting in its own right.

Minor comments:

page 2, line 102 "explicit the concentration"

page 3, limne 129, I believe it should be dx

page 4, section 4, {\cal P} has been used both to denote family of distributions and problem

Summary: The paper looks technically correct, but the techniques used for analysis of TS are not novel.

Submitted by Assigned_Reviewer_4

The paper provides a logarithmic regret bound for a known bandit algorithm, Thompson Sampling, in the case of bandit problems with reward distributions being part of the exponential family. The result is novel and generalizes previous work done for the case of Bernoulli bandits.

Quality:
The paper seems technically sound. The technical results are interesting and the proofs are non-trivial.

One key difference with previous work is the change of the prior distribution for Thompson Sampling (from a uniform prior to Jeffreys prior). The authors provide reasons as to why the Jeffreys prior should be used since it helps the theoretical analysis. My first question is though, what would be the motivation in an applied setting to prefer a Jeffreys to a uniform prior? And what would be the practical impact of this change? For example, in the case of a Bernoulli distribution, the Jeffreys prior is Beta(0.5,0.5) which encodes a prior belief that the optimal expected reward is either closer to 0 or closer to 1. This seems unreasonable if no prior knowledge exists about the arms.

My second question is about the proof of inequality (6) in Appendix A. At line 517, D_{a,s} is defined to be the complement of the second part of the event E_{a,t} defined in line 219 (equation (5)). But D_{a,s} is not really the complement (as it would require an \forall s' quantifier instead of the existential quantifier). I am assuming the authors skipped some steps, so could you please clarify?

Clarity:
The paper is relatively difficult to read. One problem is the disproportionate use of mathematical notation at the expense of providing intuitions for the used variables and the results. Here are several examples:

- L(\theta) (introduced in line 216), while an important variable for the proof, has no explanation as to why it is defined that way other than it is a "likely realization of arm a".
- The superscript 'c' used for the first time in line 244 and used through the proof has no explanation - the reader is forced to infer that the authors mean the complement of the event E_{a,t}.
- The quantity t^b introduced in line 247 is not defined and it is only introduced later in the text.
- A lot of subscripts and superscripts (like TS, pi_J) could easily be removed with the benefit of improving the readability of the results.

Regarding the organization, I like that the authors provide the high level argument "graph" before the proofs, but I think the paper could be improved if this high level description would be more intuitive even at the cost of moving more technical details to the appendix.

Another concrete example of what could help the readability is a corollary of Theorem 4 that would make it explicit for a particular distribution class.

Originality and Significance:
I agree with the authors that Theorem 4, that introduces a new tool for proving concentration bounds for posterior distributions, is new (to the best of my knowledge) and interesting in itself.

The high level strategy of the proof for Theorem 2 (the main result of the paper) follows the idea of the proof for the Bernoulli bandit case in Kaufman et al 2012 while being more technically involved and general.
Summary: While the clarity of the write-up should be improved, the paper is an interesting contribution to the literature both from the perspective of providing new technical tools for solving bandit problems and for understanding the properties of Thompson sampling.

Submitted by Assigned_Reviewer_5

Previous work showed that for bandit problems with binary reward, Thompson Sampling achieves the asymptotic lower bound on regret of Lai and Robbins if it is applied with a Beta prior. This paper shows this lower bound is also attained when rewards distributions are in the 1-dimensional family and the algorithm uses Jeffry's prior.

The paper is clearly written and organized. Its strength rests on a single powerful theorem. This main result is strong enough to merit acceptance almost on its own. Still, the paper needs to go further to be truly great. The proof is quite technical, and doesn't seem to provide clear new insight into Thompson Sampling. The result relies heavily on the use of an uniformative prior and it feels like an asymptotic guarantee does not require such a strong restriction. Still, the main result is impressive, and the paper deserves to be accepted.
Summary: This paper merits acceptance due to the strength of its main theorem.
Author Feedback

Author rebuttal: We thank all the reviewers for there careful and thoughtful reading of our work.

We thank Reviewer 3 for pointing out our omission of Algorithm 2 from [1] in the introductory discussion of our submission. The reviewer is correct that finite-time bounds do exist for bounded distributions through a small modification of the TS algorithm for Bernoulli models, and we will clarify this in the paper. However, there is no proven lower bound for general bounded rewards, and the main contribution of our paper is precisely to prove optimality (i.e. matching Lai and Robbins' bound) of the vanilla TS algorithm for more general distributions than Bernoulli, including unbounded ones (as mentioned by Reviewer 3), which is new.

Both Reviewer 4 and Reviewer 5 have commented on the readability of the paper, and we will make every effort to improve it, particularly by giving more intuition on the quantities we introduce, and removing superfluous superscripts and subscripts.

Reviewer 4 also asked two specific questions. The first is about the practical motivation for the use of Jeffreys priors over uniform priors. To measure the practical impact of the change of prior between uniform and Jeffreys, numerical comparison on various problems would also be interesting. For instance in Bernoulli bandit problems with small rewards, Jeffreys prior might be better. In general, this is an old and undecided debate, however one usual response is that uniform (or flat) priors are not invariant under reparametrisation of the parameter space. As an example, consider the uniform prior for a Bernoulli model, i.e., the uniform prior on the expected value of success, \mu (call it u = u(\mu)). Now suppose that we reparametrise the parameter space, and use instead the natural parameter

\theta = \log(\mu/(1 - \mu));

then the uniform prior $u$ is transformed to the prior on theta

u(\theta) = e^{\theta}/(1+ e^{\theta})^2,

which is no longer flat. By contrast the Jeffrey's prior distribution is invariant under reparametrisation of the space, and so is in this sense more uninformative than the uniform distribution. We refer the reader to Chapter 3.5 of the book The Bayesian Choice by Christian P Robert for a more detailed discussion.

The second question was relative to the decomposition on line 512 in the appendix. Indeed several steps were omitted here which we will include. We have used the fact that for any events {B_s, C_s}_{s = 1,...,t} we have the following set inclusion (please put directly into latex to read it more clearly):

\left(\bigcup_{s=1}^t B_s \cap C_s\right)^c = \bigcup_{s=1}^t \left(B_s^c \cup C_s^c\right)
\subset \bigcup_{s=1}^t \left(B_s^c \cup \bigcup_s C_s^c\right)
= \left(\bigcup_{s=1}^t B_s^c\right) \cup \left(\bigcup_s C_s^c\right).